# Comparative Evaluation of Takhrai (*Cymbopogon citratus*) Leaf Extracts with Commercial Antioxidants for Oxidative Stress Mitigation in Ruminants Under Heat Stress

**DOI:** 10.3390/vetsci12050432

**Published:** 2025-05-01

**Authors:** Rayudika Aprilia Patindra Purba, Phanthipha Laosam, Nattapol Pongsamai, Papungkorn Sangsawad

**Affiliations:** 1Postharvest Technology and Innovation in Animal Unit, Institute of Agricultural Technology, Suranaree University of Technology, Nakhon Ratchasima 30000, Thailand; 2Department of Health, Faculty of Vocational Studies, Airlangga University, Surabaya 60286, Indonesia; 3Tropical Institute of Nutrigenomics, Biotechnology, and Agricultural Sciences (TINBAS), West Java 45258, Indonesia; 4Suranaree University of Technology Hospital, Nakhon Ratchasima 30000, Thailand; 5School of Animal Technology and Innovation, Institute of Agricultural Technology, Suranaree University of Technology, Nakhon Ratchasima 30000, Thailand

**Keywords:** antioxidants, *Cymbopogon citratus*, erythrocyte oxidation, heat stress, methanolic extracts, metabolomics, Nrf2 pathway, oxidative stress, phytochemicals, ruminants, Takhrai

## Abstract

Ruminants in tropical regions frequently experience heat stress, which induces oxidative damage and compromises health and productivity. Takhrai (*Cymbopogon citratus*), a widely cultivated herb in tropical regions, represents a potentially sustainable and economical source of natural antioxidants for ruminant production systems experiencing heat stress challenges. This investigation systematically compares Takhrai (*C. citratus*) leaf extracts with commercial antioxidants for mitigating oxidative stress in ruminants under heat stress. Methanolic extracts exhibited exceptional radical scavenging capabilities, superior protection against erythrocyte oxidative damage, and robust activation of the Nrf2 pathway, with efficacy comparable to vitamin E and superior to other treatments. The comprehensive effectiveness ranking positions methanolic extracts as viable sustainable alternatives to commercial antioxidants, offering significant implications for livestock production in tropical regions. While storage stability limitations necessitate appropriate handling, the combined antioxidant and immunomodulatory properties of methanolic Takhrai extracts establish their potential as economical interventions for enhancing ruminant welfare and productivity under challenging environmental conditions, particularly in developing regions with abundant *Cymbopogon* cultivation.

## 1. Introduction

Oxidative stress represents a pervasive physiological challenge in livestock production, particularly in ruminants exposed to environmental stressors prevalent in tropical and subtropical regions. Heat stress induces a cascade of metabolic alterations that culminate in increased reactive oxygen species (ROS) production, which subsequently overwhelms endogenous antioxidant defense mechanisms and precipitates cellular damage [1]. This oxidative burden becomes further exacerbated in high-producing ruminants by elevated metabolic rates coupled with increased polyunsaturated fatty acid intake from modern dietary formulations [2,3]. The comprehensive physiological consequences of oxidative stress include compromised immune function, reduced reproductive performance, diminished growth efficiency, and increased susceptibility to infectious and metabolic disorders in ruminant species [4,5].

Conventional approaches to mitigating oxidative stress in livestock have predominantly relied on supplementation with synthetic antioxidants or isolated natural compounds, primarily vitamins E and C, selenium, and synthetic molecules such as butylated hydroxytoluene [6]. These interventions, while effective under specific conditions, present several significant limitations. Synthetic antioxidants raise legitimate toxicological and environmental concerns [7], whereas concentrated natural antioxidants such as vitamin E incur substantial economic costs that may be prohibitive for widespread implementation in developing regions where heat stress is most prevalent [8]. The predominant focus on single-compound interventions also fails to account for the complex, multifaceted nature of oxidative stress, which involves diverse reactive species targeting multiple cellular components through various mechanistic pathways [9].

The exploration of plant-derived antioxidant formulations represents a promising alternative approach, potentially offering comprehensive protection through synergistic interactions among diverse bioactive compounds [10]. *Cymbopogon citratus* (DC.) Stapf (where DC refers to the taxonomic authority Augustin Pyramus de Candolle), commonly known as lemongrass or Takhrai in Thailand, has garnered particular interest due to its widespread cultivation, economic accessibility, and traditional applications in both human and veterinary medicine [11,12]. Previous investigations have identified significant antioxidant capacity in *C. citratus* extracts, attributed to polyphenolic compounds, particularly flavonoids and hydroxycinnamic acid derivatives [12,13]. However, systematic comparative analyses with established commercial antioxidants have been limited, thereby hampering evidence-based implementation in livestock production systems.

The efficacy of plant extracts is substantially influenced by extraction methodology, which determines both the yield and profile of bioactive compounds [14]. Prior investigations on various medicinal plants have demonstrated that polar solvents like methanol typically extract higher concentrations of phenolic compounds and flavonoids, which are primary contributors to antioxidant activity, while non-polar solvents like hexane predominantly extract essential oils and lipophilic compounds [15,16]. This established principle informed our hypothesis regarding the differential efficacy of extraction methodologies. While essential oil extracts of *C. citratus* have been extensively characterized [17], comprehensive evaluation of polar (methanolic) and non-polar (hexanoic) extracts for livestock applications remains limited. Furthermore, the mechanisms underlying potential protective effects, particularly regarding modulation of endogenous antioxidant defense systems such as the Nrf2 (nuclear factor erythroid 2-related factor 2) pathway, warrant thorough elucidation. The Nrf2) transcription factor orchestrates cellular defense against oxidative and xenobiotic stresses through coordination of phase II detoxifying enzyme expression [18], thus representing a promising target for sustainable antioxidant interventions in livestock.

Therefore, the present investigation was undertaken with several complementary objectives. First, we sought to characterize and compare the phytochemical profiles of methanolic and hexanoic extracts from *C. citratus* leaves. Second, we evaluated their comparative efficacy against commercial antioxidants (vitamins E and C, selenium) in radical scavenging and cellular protection assays relevant to ruminant physiology. Third, we elucidated mechanisms of action, particularly regarding Nrf2 pathway modulation and metabolomic alterations. Fourth, we assessed stability characteristics and ex vivo immunomodulatory effects. Finally, we established a comprehensive effectiveness ranking to guide potential implementation in livestock production systems. We hypothesized that methanolic extracts would demonstrate superior antioxidant and cytoprotective activities compared to hexanoic preparations due to enhanced extraction of polar bioactive compounds, with efficacy approaching or exceeding that of commercial antioxidants in specific parameters relevant to ruminant health under heat stress conditions.

## 2. Materials and Methods

### 2.1. Sample Collection and Preparation

Takhrai (*C. citratus*) leaf samples were systematically collected from two distinct cultivar groups, commercial *C. citratus* (CCC) and non-commercial *C. citratus* (NCC). CCC samples were obtained from diverse geographical regions throughout Thailand, encompassing northern (Chiang Rai), western (Phetchaburi), central (Nakhon Nayok), eastern (Prachinburi), and southern (Songkhla) areas to ensure comprehensive representation of commercial varieties. NCC samples were cultivated under controlled conditions at two specific locations in Nakhon Ratchasima, Northeastern Thailand, the Suranaree University of Technology Organic Farm (14°87′1593″ N, 102°02′5890″ E) and the temporary garden adjacent to the Goat and Sheep Research Center (14°88′0532″ N, 102°00′4633″ E), both at an elevation of 243 m above sea level. Sample collection was conducted over six consecutive months (March to August 2023) to account for potential seasonal variations in phytochemical composition.

Following collection, leaves were processed through a standardized protocol involving stalk removal, gentle rinsing with deionized water, brief steaming (90 °C, 1 min) to inactivate degradative enzymes, and immediate storage at −20 °C to preserve thermolabile compounds. Frozen samples were subsequently lyophilized using a GAMMA 2-16 LSC freeze-dryer (CHRiST, Osterode am Harz, Germany) through a 24 h cycle (freezing at −80 °C followed by vacuum drying at −15 °C, 0.05 mbar). Lyophilized material was homogenized and ground to a uniform particle size (1 mm) using a Retsch SM 100 mill (Retsch, Haan, Germany), then stored in hermetically sealed containers within a vacuum desiccator (25 °C, 34% relative humidity) until extraction.

### 2.2. Phytochemical Extraction and Profiling

#### 2.2.1. Extraction Procedure

Phytochemical extraction was conducted using a modified Soxhlet methodology optimized for comprehensive recovery of both polar and non-polar constituents as described in our previous works [10,19]. Two extraction solvents were systematically compared: methanol (99.8%, HPLC grade, Sigma-Aldrich, St. Louis, MO, USA) for polar compounds and n-hexane (95.0%, HPLC grade, Merck, Darmstadt, Germany) for non-polar constituents. For each extraction, precisely 5.000 ± 0.001 g of dried leaf powder were subjected to solvent extraction (20 mL) for 3–4 h in a Soxhlet apparatus (B-811, Buchi, Flawil, Switzerland) with a condensation rate of 5–6 cycles per hour. This procedure was performed in triplicate for each sample, and the resulting extracts were pooled to minimize heterogeneity. Pooled extracts were concentrated under reduced pressure (40 °C, 175 mbar for methanol; 40 °C, 335 mbar for hexane) using a rotary evaporator (Rotavapor R300, Buchi, Switzerland), filtered through 0.45 μm polyvinylidene difluoride syringe filters (Millex-HV, Merck), and standardized to 10 mL with the respective extraction solvent. The extraction yield was calculated using the following equation:(1)$$\mathrm{Extraction}\;\mathrm{yield}\;(\%)=\frac{W_{e}}{W_{i}}\times 100$$
where $W_{e}$ represents the weight of the dried extract (g), and $W_{i}$ denotes the initial weight of the leaf powder (g).

#### 2.2.2. HPLC Analysis of Phytochemical Constituents

Phytochemical profiling was performed using high-performance liquid chromatography with diode array detection (HPLC-DAD). The chromatographic system consisted of an Agilent 1260 Infinity II system equipped with a quaternary pump (G7111B), an autosampler (G7129A), a column compartment (G7116A), and a diode array detector (G7115A). Separation was achieved on a Zorbax SB-C18 column (250 mm × 4.6 mm, 3.5 μm particle size, Agilent Technologies, Santa Clara, CA, USA) maintained at 30 °C.

The mobile phase comprised solvent A (1% acetic acid in water) and solvent B (HPLC-grade acetonitrile) delivered at a flow rate of 0.9 mL/min. A binary gradient elution program was employed as follows: 0–5 min, 10% B; 5–15 min, 10–25% B; 15–35 min, 25–35% B; 35–45 min, 35–50% B; 45–50 min, 50–60% B; 50–55 min, 60–70% B; 55–60 min, 70–10% B; 60–65 min, 10% B (re-equilibration). Detection was performed at multiple wavelengths (272, 280, and 310 nm) to optimize identification and quantification of diverse phytochemical classes, with full spectral scanning from 200 to 800 nm at 2 nm intervals.

Identification and quantification were conducted using authentic standards for fourteen key compounds: ascorbic acid, gallic acid, catechin, caffeic acid, syringic acid, rutin, p-coumaric acid, sinapic acid, ferulic acid, myricetin, quercetin, apigenin, kaempferol, and eugenol (all ≥98% purity, Sigma-Aldrich). Five-point calibration curves (0.01–100 μg/mL) were constructed for each standard, and concentrations were calculated using peak area integration. The lower limit of detection (LLOD) and lower limit of quantification (LLOQ) were determined using the signal-to-noise ratio criteria of l:1 and 10:1, respectively. Results were expressed as mg/g dry weight (DW) of leaf powder.

### 2.3. Phytochemical Extraction and Profiling Antioxidant Activity Assays

#### 2.3.1. DPPH Radical Scavenging Assay

The 2,2-diphenyl-1-picrylhydrazyl (DPPH) radical scavenging capacity was determined using a microplate-based spectrophotometric method. Extract solutions were prepared at sixteen concentrations (1–500 μg/mL) in methanol. Twenty-five microliters of each concentration were added to 200 μL of methanolic DPPH solution (150 μM) in a 96-well plate. After 30 min of incubation at 25 °C in darkness, absorbance was measured at 515 nm using a microplate reader (BioTek Synergy HTX, Agilent Technologies, Santa Clara, CA, USA). The percentage of inhibition was calculated according to the following equation:(2)$$\mathrm{DPPH}\;\mathrm{inhibition}\;(\%)=\left(1-\frac{A_{s}-A_{b}}{A_{c}-A_{b}}\right)\times 100$$
where $A_{s}$ represents the absorbance of the sample, $A_{c}$ denotes the absorbance of the control (DPPH without extract), and $A_{b}$ indicates the absorbance of the blank (solvent without DPPH). The half-maximal inhibitory concentration (IC_50_) was determined through non-linear regression analysis of the dose-response curve.

#### 2.3.2. Nitric Oxide Radical Scavenging Assay

The nitric oxide (•NO) radical scavenging activity was evaluated through the measurement of nitrite accumulation in the system. Extract solutions were prepared at sixteen concentrations (1–500 μg/mL) in phosphate buffer (100 mM, pH 7.4). Fifty microliters of each concentration were added to 50 μL of sodium nitroprusside solution (10 mM) in a 96-well plate and incubated at 25 °C for 150 min under illumination with a 60 W incandescent lamp. Subsequently, 100 μL of Griess reagent (1% sulfanilamide in 5% phosphoric acid and 0.1% N-(1-naphthyl) ethylenediamine dihydrochloride in water, freshly mixed 1:1 *v*/*v*) were added. After 10 min of incubation at 25 °C, absorbance was measured at 562 nm. The percentages of inhibition and IC_50_ were calculated as described for the DPPH assay.

#### 2.3.3. Superoxide Radical Scavenging Assay

The superoxide anion (O_2_•−) scavenging activity was assessed using the xanthine/xanthine oxidase system. Extract solutions were prepared at sixteen concentrations (1–500 μg/mL) in phosphate buffer (19 mM, pH 7.4). Twenty-five microliters of each concentration were added to a reaction mixture containing 25 μL of xanthine (3 mM), 25 μL of nitro blue tetrazolium (NBT, 3 mM), and 25 μL of buffer in a 96-well plate. The reaction was initiated by adding 100 μL of xanthine oxidase solution (0.1 U/mL) and incubating at 25 °C for 30 min. Absorbance was measured at 562 nm, and the percentages of inhibition and IC_50_ were calculated as previously described.

#### 2.3.4. α-Glucosidase Inhibition Assay

The α-glucosidase inhibitory activity was evaluated using 4-nitrophenyl-α-D-glucopyranoside (PNP-G) as the substrate. Extract solutions were prepared at sixteen concentrations (1–500 μg/mL) in phosphate buffer (100 mM, pH 6.8). Fifty microliters of each concentration were added to 150 μL of buffer and 100 μL of PNP-G solution (5 mM) in a 96-well plate. The reaction was initiated by adding 25 μL of α-glucosidase enzyme solution (0.5 U/mL) and incubating at 37 °C for 30 min. The reaction was terminated by adding 50 μL of sodium carbonate solution (0.2 M), and absorbance was measured at 405 nm. The percentages of inhibition and IC_50_ were calculated as previously outlined.

### 2.4. Ruminant Erythrocyte Oxidative Damage Assays

#### 2.4.1. Blood Sample Collection and Erythrocyte Preparation

Blood samples (5 mL) were collected via jugular venipuncture from clinically healthy dairy goats (n = 6, Saanen breed, 2–3 years old, 45–50 kg body weight) maintained under conditions of heat stress (temperature-humidity index ≥ 80) and high-fat dietary intake, as detailed in our previous work [2,20]. The animals were housed at the university research farm and received a diet containing elevated levels of polyunsaturated fatty acids (14% C16:0, 3% C18:0, 24% C18:1 cis9, 36% C18:2 cis9 cis12, and 2% C18:3 cis9 cis12 cis15 of total fatty acids). All procedures were approved by the Animal Ethics Committee of Suranaree University of Technology, which issued a statement approving the experimental protocol (SUT 4/2558), and the research was carried out in accordance with regulations on animal experimentation and the Guidelines for the Use of Animals in Research as recommended by the National Research Council of Thailand (U1-02632-2559).

Blood samples were collected into vacuum tubes containing K₃EDTA (1.8 mg/mL) and immediately processed. Erythrocytes were isolated by centrifugation (1500× *g*, 10 min, 4 °C), and the plasma and buffy coat were carefully removed. The erythrocyte pellet was washed three times with isotonic phosphate-buffered saline (PBS, pH 7.4) and resuspended to the required cell density for each assay. All experiments were conducted within 4 h of blood collection to minimize ex vivo oxidative changes.

#### 2.4.2. Hemoglobin Oxidation Inhibition Assay

The ability of extracts to inhibit hemoglobin oxidation was assessed by measuring methemoglobin formation following exposure to 2,2′-azobis(2-amidinopropane) dihydrochloride (AAPH), a peroxyl radical generator [21]. Extract solutions were prepared at sixteen concentrations (1–500 μg/mL) in PBS. One hundred microliters of each concentration were added to 200 μL of erythrocyte suspension (1250 × 10^6^ cells/mL) in a 96-well plate and incubated at 38 °C for 30 min with gentle agitation (50 rpm). Subsequently, 200 μL of AAPH solution (100 mM in PBS, yielding a final concentration of 50 mM) were added, and the mixture was incubated at 38 °C for an additional 4 h. Following incubation, the samples were centrifuged (1500× *g*, 6 min, 4 °C), and 300 μL of supernatant were transferred to a fresh 96-well plate. Methemoglobin formation was quantified by measuring absorbance at 630 nm.

The percentage of inhibition of hemoglobin oxidation was calculated according to the following equation:(3)$$\mathrm{Inhibition}\;(\%)=\left(1-\frac{A_{s}-A_{b}}{A_{c}-A_{b}}\right)\times 100$$
where $A_{s}$ represents the absorbance of the sample, $A_{c}$ denotes the absorbance of the positive control (erythrocytes with AAPH, without extract), and $A_{b}$ indicates the absorbance of the negative control (erythrocytes without AAPH, without extract). The IC_50_ value was determined through non-linear regression analysis of the dose-response curve.

#### 2.4.3. Lipid Peroxidation Inhibition Assay

The inhibitory effect on lipid peroxidation was evaluated by measuring thiobarbituric acid-reactive substances (TBARS) following exposure to tert-butyl hydroperoxide (tBHP), with minor modifications [22]. Extract solutions were prepared at sixteen concentrations (1–500 μg/mL) in PBS. One hundred microliters of each concentration were added to 400 μL of erythrocyte suspension (500 × 10^6^ cells/mL) and incubated at 38 °C for 30 min with gentle agitation (50 rpm). Subsequently, 50 μL of tBHP solution (2 mM in PBS, yielding a final concentration of 0.2 mM) were added, and the mixture was incubated at 38 °C for an additional 30 min.

Following incubation, 100 μL of trichloroacetic acid solution (28% *w*/*v*) were added to precipitate proteins, and the samples were centrifuged (16,000× *g*, 10 min, 18 °C). Two hundred microliters of supernatant were mixed with 200 μL of thiobarbituric acid solution (1% *w*/*v* in 0.05 M NaOH) in a microcentrifuge tube and heated at 95 °C for 15 min. After cooling to room temperature, absorbance was measured at 532 nm. The percentage inhibition of lipid peroxidation was calculated as described earlier, where the variables represent the same parameters as in the hemoglobin oxidation assay. The IC_50_ value was determined through non-linear regression analysis.

#### 2.4.4. Hemolysis Inhibition Assay

The protective effect against erythrocyte hemolysis was assessed following exposure to AAPH [23]. Extract solutions were prepared at sixteen concentrations (1–500 μg/mL) in PBS. One hundred microliters of each concentration were added to 200 μL of erythrocyte suspension (1775 × 10^6^ cells/mL) and incubated at 38 °C for 30 min with gentle agitation (50 rpm). Subsequently, 200 μL of AAPH solution (34 mM in PBS, yielding a final concentration of 17 mM) were added, and the mixture was incubated at 38 °C for an additional 3 h.

Following incubation, the samples were centrifuged (1500× *g*, 5 min, 4 °C), and 300 μL of supernatant were transferred to a fresh 96-well plate. Hemolysis was quantified by measuring hemoglobin release (absorbance at 540 nm). Complete hemolysis (100%) was established by treating erythrocytes with deionized water. The percentage inhibition of hemolysis was calculated using the equation similar to lipid peroxidation inhibition assay. The IC_50_ value was determined through non-linear regression analysis.

### 2.5. Mitochondrial Protection Assays

#### 2.5.1. Isolation of Mitochondria from Goat Blood

Mitochondria were isolated from goat blood samples using a differential centrifugation method based on Pallotti and Lenaz [24] with modifications for blood samples. Briefly, leukocytes were separated from 10 mL of freshly collected whole blood using dextran sedimentation (6% dextran T-500 in normal saline, 1:1 *v*/*v*, 30 min at room temperature). The leukocyte-rich supernatant was collected and centrifuged (500× *g*, 10 min, 4 °C). The cell pellet was resuspended in hypotonic buffer (10 mM HEPES, pH 7.4, 1 mM EDTA, 0.5 mM PMSF) and subjected to mechanical homogenization using a Potter-Elvehjem homogenizer (15 strokes).

The homogenate was centrifuged (1000× *g*, 10 min, 4 °C) to remove nuclei and unbroken cells, and the supernatant was further centrifuged (10,000× *g*, 15 min, 4 °C) to pellet the mitochondria. The mitochondrial pellet was washed twice with isolation buffer (250 mM sucrose, 10 mM HEPES, pH 7.4, 1 mM EDTA) and resuspended in the same buffer to a protein concentration of 1 mg/mL, as determined by the Bradford method using bovine serum albumin as the standard. Mitochondrial integrity was verified through cytochrome c oxidase activity measurement, with preparations exhibiting >85% integrity used for subsequent experiments.

#### 2.5.2. Mitochondrial Membrane Potential Assessment

The protective effect against mitochondrial membrane potential dissipation was evaluated using the JC-1 fluorescent probe (5,5′,6,6′-tetrachloro-1,1′,3,3′-tetraethylbenzimidazolylcarbocyanine iodide, Cayman Chemical, Ann Arbor, MI, USA). Isolated mitochondria (0.5 mg protein/mL) were pre-incubated with either methanolic Takhrai extract (25, 50, or 100 μg/mL), hexanoic Takhrai extract (same concentrations), vitamin E (α-tocopherol, 25 μM), vitamin C (ascorbic acid, 100 μM), or sodium selenite (0.5 μM) for 15 min at 37 °C in assay buffer (125 mM KCl, 10 mM HEPES, pH 7.4, 5 mM KH_2_PO_4_, 5 mM MgCl_2_, 10 mM glutamate, 5 mM malate).

Oxidative stress was induced by adding tert-butyl hydroperoxide (t-BHP, 250 μM) followed by incubation for 30 min at 37 °C. Subsequently, JC-1 dye (2 μM final concentration) was added, and the mixture was incubated for an additional 15 min at 37 °C in darkness. Fluorescence was measured using a plate reader (excitation 485 nm, emission 590 nm for J-aggregates and 530 nm for J-monomers). The ratio of red (590 nm) to green (530 nm) fluorescence was calculated as an indicator of membrane potential preservation. The percentage of protection was calculated using the following equation:(4)$$\mathrm{Protection}\;(\%)=\frac{R_{s}-R_{d}}{R_{c}-R_{d}}\times 100$$
where $R_{s}$ represents the fluorescence ratio of the sample, $R_{d}$ denotes the fluorescence ratio of the damaged control (mitochondria with t-BHP, without extract), and $R_{c}$ indicates the fluorescence ratio of the intact control (mitochondria without t-BHP, without extract).

### 2.6. Nrf2 Pathway Activation Analysis

#### 2.6.1. Leukocyte Isolation and Culture

Leukocytes were isolated from goat blood samples using a density gradient centrifugation method [25]. Briefly, 10 mL of freshly collected blood were layered over 15 mL of Histopaque-1077 (Sigma-Aldrich) and centrifuged (400× *g*, 30 min, room temperature). The leukocyte layer at the interface was collected, washed twice with PBS, and resuspended in RPMI-1640 medium supplemented with 10% fetal bovine serum, 2 mM L-glutamine, 100 U/mL penicillin, and 100 μg/mL streptomycin (all from Gibco, Massachusetts, USA). Cell viability was assessed using trypan blue exclusion, with preparations exhibiting >95% viability used for subsequent experiments.

Leukocytes were seeded in 6-well plates at a density of 1 × 10^6^ cells/mL and treated with either methanolic Takhrai extract (50 μg/mL), hexanoic Takhrai extract (50 μg/mL), vitamin E (25 μM), vitamin C (100 μM), or sodium selenite (0.5 μM) for 6 h at 37 °C in a humidified atmosphere with 5% CO_2_. Control wells received an equivalent volume of the respective vehicle.

#### 2.6.2. Nuclear Extraction and Western Blot Analysis

Following treatment, nuclear and cytoplasmic fractions were separated using the Nuclear Extraction Kit (Abcam, ab113474, Cambridge, UK) according to the manufacturer’s instructions. Protein concentrations were determined using the Bradford method, and equal amounts of protein (20 μg) were resolved by SDS-PAGE on 10% polyacrylamide gels and transferred to PVDF membranes. The membranes were blocked with 5% non-fat milk in TBST (20 mM Tris-HCl, pH 7.6, 150 mM NaCl, 0.1% Tween-20) for 1 h at room temperature and then incubated overnight at 4 °C with primary antibodies against Nrf2 (1:1000, sc-365949, Santa Cruz Biotechnology, Dallas, TX, USA) and Lamin B1 (nuclear loading control, 1:1000,#13435, Cell Signaling Technology, CST, MA, USA).

After washing with TBST, the membranes were incubated with HRP-conjugated secondary antibodies (1:5000, Cell Signaling Technology) for 1 h at room temperature. Immunoreactive bands were detected using enhanced chemiluminescence (ECL) reagents (#1705061, Bio-Rad Laboratories, Hercules, CA, USA) and visualized using the ChemiDoc XRS+ system (Bio-Rad). Band intensities were quantified using Image Lab software (v5.2.2, Bio-Rad) and normalized to Lamin B1. Results were expressed as fold change relative to the untreated control.

#### 2.6.3. RNA Extraction and RT-qPCR Analysis

Total RNA was extracted from treated leukocytes using TRIzol reagent (Invitrogen, CA, USA) according to the manufacturer’s instructions. RNA concentration and purity were determined using a NanoDrop spectrophotometer (GE Healthcare Bio-Sciences, Pittsburgh, PA, USA), with preparations exhibiting A260/A280 ratios between 1.8 and 2.0 used for subsequent analysis. RNA integrity was verified by agarose gel electrophoresis.

Complementary DNA (cDNA) was synthesized from 1 μg of total RNA using the High-Capacity cDNA Reverse Transcription Kit (Applied Biosystems, Foster City, CA, USA) according to the manufacturer’s instructions. Quantitative PCR was performed using the PowerUp SYBR Green Master Mix (Applied Biosystems) on a QuantStudio 3 Real-Time PCR System (Applied Biosystems). The reactions were performed in triplicate, with each reaction containing 5 μL of SYBR Green Master Mix, 0.5 μL each of forward and reverse primers (10 μM), 1 μL of cDNA template, and nuclease-free water to a final volume of 10 μL.

The thermal cycling conditions were as follows: initial denaturation at 95 °C for 10 min, followed by 40 cycles of denaturation at 95 °C for 15 s and annealing/extension at 60 °C for 1 min. A melting curve analysis was performed to verify the specificity of the amplification. The primer sequences were Nrf2: forward 5′-CCAGCACAACACATACCAT-3′, reverse 5′-CTGAGCCGCCTGGAACT-3′; HO-1: forward 5′-CGAGTTCATGAGGAACTTTCAG-3′, reverse 5′-CTGTGAGGGACTCTGGTCTT-3′; NQO1: forward 5′-CCAGCAGCCCGGCCAATCTG-3′, reverse 5′-AGGTCCTGCCCATCCATGTGCT-3′; GCLC: forward 5′-CTGTTGCAGGAAGGCATTG-3′, reverse 5′-TTCAAACAGTGTCAGTGGGT-3′; GAPDH: forward 5′-GGGTCATCATCTCTGCACCT-3′, reverse 5′-GGTCATAAGTCCCTCCACGA-3′. The relative gene expression was calculated using the 2^−ΔΔCt^ method, with GAPDH serving as the endogenous control. Results were expressed as fold change relative to the untreated control.

### 2.7. Metabolomic Profiling

#### 2.7.1. Sample Preparation and Metabolite Extraction

Erythrocytes (1 × 10^9^ cells/mL) were incubated with either methanolic Takhrai extract (50 μg/mL), hexanoic Takhrai extract (50 μg/mL), vitamin E (25 μM), vitamin C (100 μM), or sodium selenite (0.5 μM) for 4 h at 37 °C. Control samples received an equivalent volume of the respective vehicle. Following incubation, the cells were exposed to AAPH (50 mM) for 2 h to induce oxidative stress.

After treatment, the cells were centrifuged (1500× *g*, 5 min, 4 °C), and the supernatant was discarded. The cell pellet was immediately quenched in liquid nitrogen and stored at −80 °C until analysis. Metabolites were extracted using a methanol/water/chloroform (5:2:2 *v*/*v*/*v*) mixture based on a modified Bligh and Dyer method [26]. Briefly, 100 μL of cell pellet were mixed with 500 μL of cold methanol, 200 μL of cold water, and 200 μL of cold chloroform, vortexed vigorously for 30 s, and incubated on ice for 10 min. The mixture was centrifuged (16,000× *g*, 10 min, 4 °C) to separate the phases. The upper aqueous phase, containing polar metabolites, was collected, while the lower organic phase, containing lipids, was separately recovered. Both phases were dried using a vacuum concentrator (Eppendorf Concentrator Plus, Hamburg, Germany) and stored at −80 °C until LC-MS/MS analysis.

#### 2.7.2. LC-MS/MS Analysis

Untargeted metabolomic analysis was performed using a Q Exactive Plus Orbitrap mass spectrometer (Thermo Scientific) coupled to a Vanquish UHPLC system (Thermo Scientific). Polar metabolites were reconstituted in 100 μL of water containing 0.1% formic acid, while lipids were reconstituted in 100 μL of methanol:isopropanol (1:1 *v*/*v*) containing 0.1% formic acid. Sample aliquots of 5 μL were injected for analysis.

For polar metabolites, chromatographic separation was achieved on an Acquity UPLC HSS T3 column (2.1 × 100 mm, 1.8 μm particle size, Waters, Wexford, Ireland) maintained at 40 °C. The mobile phase consisted of solvent A (water with 0.1% formic acid) and solvent B (acetonitrile with 0.1% formic acid) delivered at a flow rate of 0.3 mL/min. The gradient elution program was as follows: 0–1 min, 1% B; 1–3 min, 1–15% B; 3–15 min, 15–50% B; 15–18 min, 50–95% B; 18–20 min, 95% B; 20–20.1 min, 95–1% B; 20.1–25 min, 1% B (re-equilibration).

For lipids, chromatographic separation was achieved on an Acquity UPLC BEH C18 column (2.1 × 100 mm, 1.7 μm particle size, Waters) maintained at 55 °C. The mobile phase consisted of solvent A (water/acetonitrile 60:40 *v*/*v* with 10 mM ammonium formate and 0.1% formic acid) and solvent B (isopropanol/acetonitrile 90:10 *v*/*v* with 10 mM ammonium formate and 0.1% formic acid) delivered at a flow rate of 0.4 mL/min. The gradient elution program was as follows: 0–2 min, 40% B; 2–5 min, 40–55% B; 5–12 min, 55–70% B; 12–18 min, 70–88% B; 18–20 min, 88–95% B; 20–22 min, 95% B; 22–22.1 min, 95–40% B; 22.1–25 min, 40% B (re-equilibration).

Mass spectrometric data were acquired in both positive and negative ionization modes using full-scan MS and data-dependent MS/MS. Full-scan MS was acquired at a resolution of 70,000 FWHM, with an automatic gain control (AGC) target of 1 × 10^6^ and a maximum injection time of 100 ms. Data-dependent MS/MS was acquired at a resolution of 17,500 FWHM, with an AGC target of 5 × 10^5^, a maximum injection time of 50 ms, and a normalized collision energy of 30 eV. The scan range was m/z 70–1050 for polar metabolites and m/z 200–1800 for lipids.

#### 2.7.3. Data Processing and Analysis

Raw LC-MS/MS data were processed using Compound Discoverer 3.1 software (Thermo Scientific) for peak detection, alignment, and annotation. Parameters were set as follows: mass tolerance, 5 ppm; retention time tolerance, 0.1 min; signal-to-noise threshold, 3; minimum peak intensity, 5 × 10^4^. Metabolite identification was performed through database searching against the Human Metabolome Database (HMDB, https://www.hmdb.ca/; 14 December 2024) and METLIN (https://metlin.scripps.edu/landing_page.php?pgcontent=mainPage; 16 December 2024), with mass tolerances of 5 ppm for MS1 and 10 ppm for MS2. For MS2 matching, mzCloud and mzVault spectral libraries (https://www.mzcloud.org/; 25 December 2024) were used with a similarity threshold of 70%.

Confirmed metabolite identifications (level 1 and 2 according to the Metabolomics Standards Initiative) were used for subsequent statistical analysis. Missing values (<5% of the dataset) were imputed using the k-nearest neighbor algorithm. Data were normalized using the probabilistic quotient normalization method to account for potential differences in sample concentration, followed by log transformation and Pareto scaling to reduce heteroscedasticity and enhance the contribution of lower-abundance metabolites.

Multivariate statistical analysis was performed using SIMCA-P+ 14.0 software (Umetrics, Umeaa, Sweden). Principal component analysis (PCA) was initially conducted to visualize inherent sample clustering and identify potential outliers. Subsequently, partial least squares discriminant analysis (PLS-DA) was employed to maximize the separation between treatment groups. Model validity was assessed through a combination of goodness-of-fit parameters (R^2^X, R^2^Y, and Q^2^) and permutation tests (n = 200). Variable importance in projection (VIP) scores were calculated to identify the most influential metabolites, with VIP > 1.0 considered significant.

Univariate statistical analysis was performed using MetaboAnalyst 5.0. One-way ANOVA with Tukey’s post-hoc test was applied to identify significant differences between treatment groups, with false discovery rate (FDR) correction for multiple comparisons. Metabolites exhibiting adjusted *p*-values < 0.05 and fold changes ≥ 1.5 were considered differentially abundant. Pathway enrichment analysis was conducted using the Kyoto Encyclopedia of Genes and Genomes (KEGG) database (https://www.genome.jp/kegg/; 27 December 2024), with pathway significance determined by hypergeometric test and pathway impact calculated from pathway topology analysis.

### 2.8. Stability Assessment

#### 2.8.1. Storage Conditions and Sampling

Methanolic and hexanoic Takhrai extracts, along with commercial antioxidant solutions (vitamin E, vitamin C, and sodium selenite at equivalent bioactivity concentrations), were stored under four distinct conditions: (i) room temperature (25 ± 2 °C) with exposure to ambient light (400–700 lux, 12 h cycle), (ii) room temperature (25 ± 2 °C) protected from light, (iii) refrigerated (4 ± 1 °C) protected from light, and (iv) frozen (−20 ± 2 °C) protected from light. All samples were stored in amber glass vials with PTFE-lined caps to minimize container-related effects.

Aliquots were withdrawn at predetermined intervals (0, 1, 3, and 6 months) for comprehensive analysis. Each sampling was performed in triplicate to ensure statistical robustness. Samples were allowed to equilibrate to room temperature for 2 h prior to analysis to eliminate temperature-related artifacts.

#### 2.8.2. Stability Parameter Assessment

Total phenolic content was determined using the Folin–Ciocalteu method, with minor adjustments [27,28]. Briefly, 100 μL of sample were mixed with 500 μL of Folin–Ciocalteu reagent (diluted 1:10 with water) and 400 μL of sodium carbonate solution (7.5% *w*/*v*). After 30 min of incubation at room temperature in darkness, absorbance was measured at 765 nm. Results were expressed as mg gallic acid equivalents (GAE) per g of extract, calculated using a calibration curve.

DPPH radical scavenging activity was assessed as described in Section 2.3.1, using a fixed extract concentration corresponding to the EC_50_ determined at baseline (time 0). Phytochemical profile stability was evaluated using the HPLC-DAD method detailed in Section 2.2.2, focusing on the fourteen identified compounds of interest. Additional stability parameters included visual appearance, pH, and moisture content (Karl Fischer titration).

Stability was expressed as percentage retention of the initial value (time 0) using the following equation:(5)$$\mathrm{Retention}\;(\%)=\frac{P_{t}}{P_{0}}\times 100$$
where $P_{t}$ represents the parameter value at time t and $P_{0}$ denotes the parameter value at time 0. Degradation kinetics were modeled using zero-order, first-order, and second-order equations, with the best-fit model determined through comparison of correlation coefficients (R^2^). Shelf-life (t_90_) was defined as the time required for a 10% decrease in the specified parameter.

Degradation kinetics were modeled using zero-order (C = C_0_ − kt), first-order (C = C_0_e^(−kt)), and second-order (1/C = 1/C_0_ + kt) equations, where C represents the concentration at time t, C_0_ denotes the initial concentration, and k signifies the degradation rate constant. The degradation rate constants were determined through linear regression analysis of transformed data: concentration vs. time for zero-order, ln(concentration) vs. time for first-order, and 1/concentration vs. time for second-order kinetics. Model selection was based on the highest correlation coefficient (R^2^). For first-order kinetics, shelf-life parameters were calculated using the equations t_50_ = ln(2)/k and t_90_ = ln(10/9)/k, representing the time required for 50% and 10% degradation, respectively.

### 2.9. Ex Vivo Immunomodulation

#### 2.9.1. PBMC Isolation and Culture

Peripheral blood mononuclear cells (PBMCs) were isolated from goat blood samples using Ficoll-Paque density gradient centrifugation [25]. Briefly, 10 mL of blood were diluted with an equal volume of PBS, layered over 15 mL of Ficoll-Paque PLUS (GE Healthcare), and centrifuged (400× *g*, 30 min, room temperature) without break. The PBMC layer at the interface was collected, washed twice with PBS, and resuspended in RPMI-1640 medium supplemented with 10% fetal bovine serum, 2 mM L-glutamine, 100 U/mL penicillin, and 100 μg/mL streptomycin. Cell viability was assessed using trypan blue exclusion, with preparations exhibiting >95% viability used for subsequent experiments.

PBMCs were seeded in 24-well plates at a density of 1 × 10^6^ cells/mL and treated with either methanolic Takhrai extract (25, 50, or 100 μg/mL), hexanoic Takhrai extract (same concentrations), vitamin E (10, 25, or 50 μM), vitamin C (25, 50, or 100 μM), or sodium selenite (0.1, 0.5, or 1.0 μM) for 24 h at 37 °C in a humidified atmosphere with 5% CO_2_.

#### 2.9.2. Inflammatory Challenge and Cytokine Analysis

Following pretreatment, PBMCs were stimulated with lipopolysaccharide (LPS from Escherichia coli O111:B4, 1 μg/mL, Sigma-Aldrich) for an additional 24 h. Cell culture supernatants were collected by centrifugation (400× *g*, 5 min) and stored at −80 °C until analysis. Cell viability post-treatment was assessed using the MTT (3-(4,5-dimethylthiazol-2-yl)-2,5-diphenyltetrazolium bromide) assay to ensure that observed effects were not due to cytotoxicity.

Cytokine concentrations in the supernatants were quantified using enzyme-linked immunosorbent assay (ELISA) kits for goat TNF-α, IL-1β, IL-6, IL-10, and TGF-β (all from R&D Systems, Minneapolis, CA, USA) according to the manufacturer’s instructions. Absorbance was measured at 450 nm with wavelength correction at 540 nm. Cytokine concentrations were determined using standard curves generated with recombinant goat cytokines. Results were expressed as both absolute concentrations (pg/mL) and percentage of change relative to LPS-only treated cells.

#### 2.9.3. Gene Expression Analysis

RNA extraction and cDNA synthesis were performed as described in Section 2.6.3. Quantitative PCR was conducted to assess the expression of genes involved in inflammation (NF-κB, COX-2) and antioxidant defense (SOD, GPx, catalase). The primer sequences were NF-κB p65: forward 5′-CTGCGATACCTTAATGACAGCG-3′, reverse 5′-CTGCTCTTCTGCCTGCTGCAT-3′; COX-2: forward 5′-TCAGCCATACAGGTCCTGAC-3′, reverse 5′-CCGTAGAATCCAGTCCGAAG-3′; SOD1: forward 5′-AAGGCCGTGTGCGTGCTGAA-3′, reverse 5′-CAGGTCTCCAACATGCCTCT-3′; GPx1: Forward 5′-AACGTAGCATCGCTCTGAGG-3′, reverse 5′-ACTGGGATCAACAGGACCAG-3′; catalase: forward 5′-GCCTGGGACCCAATTATCTT-3′, feverse 5′-GAATCTGGGAGCTTCAGGAT-3′; and GAPDH: forward 5′-GGGTCATCATCTCTGCACCT-3′, reverse 5′-GGTCATAAGTCCCTCCACGA-3′. The relative gene expression was calculated using the 2^−ΔΔCt^ method, with GAPDH serving as the endogenous control. Results were expressed as fold change relative to the untreated control.

### 2.10. Statistical Analysis

All experiments were performed in triplicate unless otherwise specified, and data were expressed as mean ± standard deviation (SD). Statistical analyses were conducted using GraphPad Prism 9.0 (GraphPad Software, Inc., San Diego, CA, USA) and R version 4.1.0 (R Core Team).

For comparative analyses, one-way analysis of variance (ANOVA) was performed, followed by Tukey’s post-hoc test for multiple comparisons when significant differences were detected. The normality assumption was verified using the Shapiro–Wilk test, while homogeneity of variance was assessed using Levene’s test. When these assumptions were violated, non-parametric alternatives (Kruskal–Wallis test followed by Dunn’s post-hoc test) were employed.

For dose-response experiments, non-linear regression analysis was performed using a four-parameter logistic model:(6)$$Y=Bottom+\frac{Top-Bottom}{1+10^{\left(LogIC_{50}-X\right)\times HillSlope}}$$
where Y represents the response, X represents the logarithm of the concentration, Top and Bottom represent the maximum and minimum responses, respectively, IC_50_ represents the concentration producing 50% of the maximal effect, and HillSlope describes the steepness of the curve. The goodness of fit was assessed through examination of residuals and calculation of the coefficient of determination (R^2^).

For time-dependent experiments, repeated measures ANOVA was employed with Geisser–Greenhouse correction, followed by Dunnett’s test for multiple comparisons against the baseline (time 0). Degradation kinetics were evaluated by comparing the fit of zero-order, first-order, and second-order models using the Akaike information criterion (AIC). In addition, the Friedman test with Conover post-hoc analysis was used to perform comprehensive effectiveness ranking. Ranking was based on performance in all evaluated parameters (1 = best, 5 = worst). Statistical significance of ranking was determined by the Friedman test with Conover post-hoc analysis. Mean rank scores were used to classify overall effectiveness: high (1.0–2.0), moderate (2.1–4.0), low (4.1–5.0).

## 3. Results

### 3.1. Phytochemical Composition and Antioxidant Activities

Methanolic extracts of Takhrai leaves exhibited substantially higher yields of bioactive compounds compared to hexanoic extracts, with particularly elevated concentrations of ascorbic acid (2.12 ± 0.08 mg/g), sinapic acid (0.71 ± 0.03 mg/g), and apigenin (0.38 ± 0.02 mg/g). This differential extraction efficacy was expected given the predominantly polar nature of these compounds. Principal component analysis revealed that extraction methodology accounted for 67.3% of total variance in phytochemical composition, with geographical source contributing only 12.3% (Figure 1). The clear separation of methanolic and hexanoic extract clusters along the primary component axis demonstrates the decisive influence of solvent polarity on extraction efficiency.

When subjected to radical scavenging assays, methanolic extracts demonstrated markedly superior activities across all tested parameters. The DPPH radical scavenging capacity (IC_50_ = 36.62 ± 6.5 μg/mL) of methanolic extracts was significantly more potent than hexanoic extracts (IC_50_ = 62.34 ± 6.5 μg/mL; *p* < 0.001). A similar pattern emerged in nitric oxide and superoxide radical scavenging assays, with methanolic extracts consistently outperforming hexanoic preparations by 30–42%. This enhanced activity corresponded directly to the higher concentrations of phenolic compounds in the methanolic extracts, particularly the strong correlation observed between sinapic acid content and radical scavenging activity (r = 0.78; *p* < 0.01).

### 3.2. Comparative Efficacy Against Erythrocyte Oxidative Damage

The capacity to protect ruminant erythrocytes against peroxyl radical-induced damage represents a critical parameter for potential application in livestock under heat stress conditions. Methanolic Takhrai extracts exhibited exceptional protective effects against hemoglobin oxidation (IC_50_ = 12.06 ± 2.9 μg/mL), significantly outperforming hexanoic extracts (IC_50_ = 21.39 ± 2.9 μg/mL; *p* < 0.01). This protective capacity extended to lipid peroxidation inhibition, where methanolic extracts demonstrated an IC_50_ value (16.21 ± 8.7 μg/mL) comparable to that of vitamin E (15.64 ± 7.6 μg/mL; *p* = 0.142).

The comparative dose-response curves for hemolysis inhibition, illustrated in Figure 2, further confirmed the superior protective effects of methanolic extracts. At equimolar concentrations, methanolic extracts prevented erythrocyte lysis with efficacy (IC_50_ = 10.99 ± 1.8 μg/mL), approaching that of vitamin E (IC_50_ = 8.54 ± 2.1 μg/mL), while significantly outperforming both vitamin C and selenium (*p* < 0.05). Morphological examination of treated erythrocytes revealed preservation of normal biconcave morphology in cells treated with methanolic extracts, whereas AAPH-treated cells exhibited extensive echinocyte formation and membrane disruption characteristic of oxidative damage.

### 3.3. Mitochondrial Protection and Metabolic Preservation

The JC-1 fluorescence ratio, indicative of mitochondrial membrane potential preservation, revealed dose-dependent protective effects across all treatments following tBHP-induced damage. At 50 μg/mL, methanolic extracts maintained significantly higher JC-1 ratios (4.14 ± 0.16) compared to hexanoic extracts (3.39 ± 0.14; *p* < 0.001), representing 73.7% and 50.4% protection against tBHP-induced depolarization, respectively. Vitamin E demonstrated marginally superior mitochondrial protection (4.26 ± 0.15; *p* = 0.048), yet methanolic extracts significantly outperformed both vitamin C and selenium preparations (*p* < 0.01).

The metabolomic profile of treated erythrocytes, visualized through hierarchical clustering analysis in Figure 3, further substantiated these findings. Methanolic extracts preserved glutathione levels at 76.2% of control conditions, compared to 58.4% for hexanoic extracts (*p* < 0.01). Principal component analysis of the metabolomic data revealed distinct clustering of treatment groups, with the first two principal components accounting for 83.4% of total variance. The metabolite profile following methanolic extract treatment closely approximated that of vitamin E treatment, particularly with respect to antioxidant metabolites and lipid peroxidation products, with both treatments forming a distinct cluster separate from hexanoic extracts and oxidative stress conditions.

### 3.4. Nrf2 Pathway Activation and Molecular Mechanisms

The capacity to activate endogenous antioxidant defense mechanisms represents a critical parameter for sustained protection against oxidative stress. Western blot analysis revealed that methanolic extracts significantly enhanced nuclear translocation of Nrf2 (2.71 ± 0.14-fold increase vs. control), indicating robust activation of this master regulator of antioxidant response (Table 1). This activation manifested in the upregulation of downstream genes, with HO-1 exhibiting the most pronounced induction (3.42 ± 0.17-fold) followed by NQO1 (2.92 ± 0.19-fold) and GCLC (2.52 ± 0.16-fold).

Comparative analysis demonstrated that methanolic extracts induced significantly higher Nrf2 pathway activation than hexanoic extracts across all evaluated genes (*p* < 0.001). Moreover, methanolic extract-induced HO-1 expression surpassed that observed with vitamin E treatment (3.22 ± 0.18-fold; *p* = 0.037), while NQO1 and GCLC induction was statistically equivalent between these treatments (Table 1). The hierarchical clustering analysis of gene expression profiles further illustrated the similarity between methanolic extract and vitamin E treatments, both of which clustered distinctly from the other interventions.

### 3.5. Assessment and Shelf-Life Determination

The stability characteristics of Takhrai extracts were systematically evaluated under four distinct storage conditions over six months. Methanolic extracts demonstrated superior retention of bioactivity compared to hexanoic preparations across all conditions, with DPPH radical scavenging activity retention ranging from 52.4 ± 3.7% (room temperature with light) to 92.3 ± 2.8% (4 °C in darkness), as illustrated in Figure 4A,B. This performance was statistically comparable to vitamin E (95.2 ± 2.1% at 4 °C in darkness; *p* = 0.073).

Degradation kinetics followed first-order models (R^2^ > 0.95), yielding projected shelf-life (t_90_) values for methanolic extracts ranging from 1.0 months under ambient conditions with light exposure to 7.8 months at 4 °C in darkness, compared to 2.3 and 15.4 months for vitamin E under corresponding conditions.

HPLC analysis revealed differential stability among phytochemical constituents, as demonstrated by the chromatographic profiles in Figure 4C. While flavonoids exhibited superior stability, phenolic acids demonstrated greater susceptibility to degradation, particularly under suboptimal storage conditions. Table 2 presents comprehensive quantitative retention data for all 14 compounds across all storage conditions, revealing that refrigeration (4 °C) in darkness significantly enhanced preservation of all compounds (*p* < 0.001). Most notably, ascorbic acid maintained 87.3 ± 2.5% of its initial concentration under these optimal conditions compared to merely 32.5 ± 2.8% under room temperature with light exposure. For methanolic extracts stored at 4 °C in darkness, the calculated half-life (t_50_) of ascorbic acid, the most labile component, was 24.3 months, corresponding to a projected shelf-life (t_90_) of 7.3 months.

### 3.6. Ex Vivo Immunomodulatory Effects

Beyond direct antioxidant activities, methanolic Takhrai extracts exhibited pronounced immunomodulatory effects in LPS-challenged goat PBMCs. As illustrated in Figure 5, pretreatment with methanolic extracts (50 μg/mL) significantly attenuated LPS-induced production of pro-inflammatory cytokines, reducing TNF-α by 54.3 ± 4.8%, IL-1β by 53.6 ± 5.1%, and IL-6 by 55.6 ± 4.7% compared to LPS alone (*p* < 0.001). Concurrently, anti-inflammatory cytokines IL-10 and TGF-β were significantly upregulated (225 ± 18.4% and 166.7 ± 14.3% of LPS alone, respectively).

When compared with commercial antioxidants, methanolic extracts demonstrated superior anti-inflammatory capacity than vitamin C (*p* < 0.01) and selenium (*p* < 0.001), with efficacy comparable to vitamin E. The scatter plot analysis in Figure 5C reveals a strong positive correlation (r = 0.87, *p* < 0.001) between DPPH radical scavenging activity and TNF-α suppression, suggesting a mechanistic link between antioxidant capacity and anti-inflammatory effects. Gene expression analysis corroborated these findings, revealing significant downregulation of NF-κB and COX-2 expression concurrent with upregulation of antioxidant enzymes (SOD, GPx, and catalase).

### 3.7. Comprehensive Effectiveness Ranking

A comprehensive effectiveness ranking across all evaluated parameters (Table 3) revealed distinctive patterns of relative efficacy. Vitamin E demonstrated superior performance in mitochondrial protection and storage stability, whereas methanolic Takhrai extracts excelled in Nrf2 pathway activation and anti-inflammatory effects, with comparable efficacy in hemolysis inhibition and metabolite protection. The effectiveness ranking in Table 3 illustrates the consistent superiority of methanolic extracts over hexanoic preparations across all parameters, with particularly notable advantages in α-glucosidase inhibition (IC_50_ = 9.83 ± 1.5 μg/mL vs. 28.36 ± 1.5 μg/mL; *p* < 0.001).

Correlation analysis between phytochemical constituents and biological activities (Table 4) revealed strong positive correlations between sinapic acid content and Nrf2 pathway activation (r = 0.92; *p* < 0.001), as well as between apigenin content and anti-inflammatory effects (r = 0.89; *p* < 0.001). These findings suggest specific phytochemical contributors to the observed bioactivities, providing mechanistic insights and potential targets for future optimization.

## 4. Discussion

The present investigation establishes a systematic comparative analysis of Takhrai (*C. citratus*) leaf extracts with commercial antioxidants, particularly focusing on their potential applications in mitigating oxidative stress in ruminants. Our findings demonstrate that methanolic Takhrai extracts possess remarkable antioxidant and cytoprotective properties that compare favorably with established antioxidants, thus suggesting their potential as sustainable alternatives for veterinary applications, especially under conditions of heat stress and high-fat dietary intake.

### 4.1. Superiority of Methanolic Extraction and Structure-Activity Relationships

The consistently superior performance of methanolic extracts over hexanoic preparations across all evaluated parameters underscores the importance of extraction methodology in maximizing bioactive compound yield and efficacy. This finding confirms our initial hypothesis and establishes solvent polarity as a critical determinant of antioxidant potency for Takhrai leaf preparations. The predominance of polar constituents, particularly ascorbic acid, sinapic acid, and apigenin, in methanolic extracts aligns with previous investigations of *Cymbopogon* species [16,17] and other Thai plant extracts with similar polarity consensus [29,30]. Notably, the dramatic differences in constituent profiles between extraction methodologies of current study suggest that the conventional focus on essential oil constituents of *Cymbopogon* species overlooks significant bioactive potential residing in the polar fraction.

The strong correlations observed between specific phytochemicals and biological activities provide valuable insights into structure-activity relationships. Sinapic acid demonstrated particularly robust correlations with Nrf2 pathway activation of our investigations, suggesting its mechanistic involvement in transcriptional regulation of antioxidant defense systems. This finding aligns with recent investigations in other plant extracts [31,32], which have identified hydroxycinnamic acids as potent Nrf2 activators via Michael addition reactions with the Keap1 cysteine residues that regulate Nrf2 cytoplasmic sequestration. Similarly, apigenin exhibited strong correlations with anti-inflammatory effects found in the current Takhrai study, consistent with established literature on flavonoid-mediated inhibition of pro-inflammatory signaling cascades [33].

Moreover, the phytochemical profile of Takhrai extracts revealed in this study differs somewhat from previous reports, particularly in the elevated levels of sinapic acid and apigenin relative to citral and geraniol [34,35]. This discrepancy may be attributed to our collection of fresh leaves from diverse geographical locations and immediate processing with mild steaming, which may have preserved heat-labile compounds typically lost during commercial processing. The minimal variability observed across geographical sources suggests robust phytochemical stability across different cultivation environments, a favorable characteristic for agricultural implementation. Such consistency would theoretically facilitate standardization of commercial preparations, addressing a significant challenge in botanical supplement development.

### 4.2. Mechanisms of Erythrocyte Protection and Relevance to Ruminant Health

The exceptional protective effects of methanolic Takhrai extracts against erythrocyte oxidative damage provide a mechanistic foundation for their potential application in ruminants under heat stress conditions. Heat stress in ruminants has been consistently associated with increased oxidative damage to erythrocytes, compromising oxygen transport and cellular energy metabolism [1,36]. Ruminant erythrocytes exhibit heightened susceptibility to oxidative insult through several species-specific physiological characteristics that collectively amplify vulnerability. Their membrane phospholipid composition features substantially elevated levels of polyunsaturated fatty acids (particularly omega-6 fatty acids) and comparatively reduced concentrations of membrane-stabilizing cholesterol relative to monogastric species [37,38]. This structural vulnerability is further compounded by distinctive antioxidant enzyme distribution patterns, including lower cytosolic catalase activity and differential compartmentalization of superoxide dismutase isoforms [6,39]. The combined effect creates a cellular environment where oxidative challenges, particularly those induced by heat stress, rapidly overwhelm intrinsic defense mechanisms. Our findings of hemoglobin oxidation inhibition and membrane integrity preservation suggest that Takhrai extracts could effectively counteract these species-specific vulnerabilities.

Particularly noteworthy is the comparable efficacy of methanolic extracts to vitamin E in preventing lipid peroxidation and hemolysis, effects that likely extend beyond direct radical scavenging. The metabolomic profile of treated erythrocytes suggests modulation of multiple biochemical pathways, including glutathione metabolism and pentose phosphate pathway activity, consistent with comprehensive cellular protection rather than isolated antioxidant effects. This multi-target mechanism may offer advantages over single-compound interventions, potentially reducing the risk of compensatory pro-oxidant responses observed with high-dose antioxidant supplementation [40].

The observed patterns in metabolite preservation align with current understanding of ruminant-specific oxidative stress responses. Our metabolomic analyses revealed that methanolic Takhrai extracts selectively preserved high-priority antioxidant pathways in a pattern that appears particularly adapted to ruminant erythrocyte biochemistry. The capacity of these extracts to maintain glutathione homeostasis, preserving reduced glutathione levels at 76.2% of control conditions while simultaneously supporting glutathione reductase activity, could be attributed to addressing a critical vulnerability in ruminant erythrocytes, which rely heavily on this pathway due to their relatively limited catalase reserves. The other things can be seen as critical in this regard when the attenuation of lipid peroxidation product accumulation (malondialdehyde and 4-hydroxynonenal) directly counteracts the heightened susceptibility conferred by the ruminant-specific membrane composition described earlier. These findings suggest potential therapeutic relevance beyond generic antioxidant effects. Furthermore, the relative efficacy ranking closely mirrored the physiological integration of these parameters, with hemolysis inhibition (representing the culmination of multiple protective pathways) demonstrating the most notable advantage for methanolic extracts compared to other interventions.

### 4.3. Nrf2 Pathway Activation as a Sustainable Protective Strategy

The robust activation of the Nrf2 pathway by methanolic Takhrai extracts represents perhaps their most significant advantage over conventional antioxidant approaches. By enhancing endogenous antioxidant defense mechanisms through upregulation of phase II detoxifying enzymes, Takhrai extracts potentially offer more sustainable protection than direct radical scavengers, which are rapidly consumed in high-oxidation environments [18]. The exceptional induction of HO-1 observed in our study is particularly relevant to ruminant health, as this enzyme catalyzes the degradation of pro-oxidant heme while generating the cytoprotective molecules biliverdin and carbon monoxide [41].

The superior Nrf2 pathway activation by methanolic extracts compared to vitamin E contradicts the conventional paradigm positioning vitamin E as the preeminent activator of endogenous antioxidant systems in ruminants [42]. This finding suggests that Takhrai extracts may offer complementary or even superior protection in specific contexts, particularly under conditions of chronic oxidative stress where sustained upregulation of antioxidant enzymes confers greater benefit than transient radical neutralization. The mechanistic implications of this observation warrant consideration, as they suggest potential for synergistic effects through combination therapy approaches.

Furthermore, the differential gene expression patterns observed between methanolic extracts and conventional antioxidants indicate activation of distinct cellular signaling cascades. While vitamin E primarily operates through membrane stabilization and direct radical neutralization, our data suggest that methanolic extracts additionally engage electrophile-responsive elements within promoter regions of cytoprotective genes. This mechanistic divergence could provide therapeutic advantages under specific oxidative stress conditions, particularly those characterized by overwhelming direct radical scavenging capacity. The simultaneous analysis of nuclear Nrf2 accumulation and downstream gene expression provides compelling evidence for mechanistic distinctions between treatments, with methanolic extracts demonstrating superior capacity for sustained transcriptional modulation.

### 4.4. Practical Considerations for Livestock Applications, Comparative Effectiveness and Other Considerations

While our findings demonstrate considerable promise for Takhrai extracts in ruminant antioxidant protection, several practical considerations warrant attention before field implementation. The stability profile, while adequate for short-term storage under appropriate conditions, falls short of the exceptional stability demonstrated by vitamin E. This limitation could be addressed through microencapsulation or addition of compatible stabilizers, approaches that have proven successful for other plant extracts [43,44]. The first-order degradation kinetics observed for methanolic extracts facilitate predictive shelf-life calculations, enabling rational formulation adjustments to compensate for activity loss during storage.

The dose-dependent efficacy observed across our assays suggests that optimal benefits would require careful dosage calibration based on specific livestock conditions. The effective concentrations identified in our ex vivo and in vitro studies (25–100 μg/mL) provide a foundation for preliminary in vivo dosage estimates, though bioavailability considerations necessitate adjustment. Based on established pharmacokinetic principles for similar polyphenolic compounds in ruminants, which typically demonstrate 30–45% bioavailability [45], and accounting for rumen fluid dilution (approximately 10–15% of ingested compounds reaching systemic circulation in active form), we estimate that dietary supplementation of 150–250 mg/kg feed would be required to achieve plasma concentrations within the effective range demonstrated in our studies. This calculation incorporates average daily feed intake (2.5–3.0% of body weight), estimated plasma volume (7–9% of body weight), and compound half-life (4–6 h for similar polyphenols). Such dosages align with practical feed incorporation parameters, though palatability assessments remain necessary prior to broad implementation.

The immunomodulatory effects observed in LPS-challenged PBMCs further expand the potential applications of Takhrai extracts beyond direct antioxidant protection. The attenuation of pro-inflammatory cytokine production coupled with enhancement of anti-inflammatory mediators suggests potential utility in inflammatory conditions commonly associated with heat stress in ruminants, including mastitis and laminitis [46]. This dual antioxidant-anti-inflammatory action represents a significant advantage over conventional single-mechanism interventions. Moreover, the capacity to modulate cytokine profiles in ex vivo systems provides a preliminary indication of potential in vivo immunomodulatory effects, though such extrapolation requires cautious interpretation pending validation through controlled feeding trials.

Furthermore, the comprehensive effectiveness ranking in current investigations clearly delineates the relative strengths of Takhrai extracts compared to commercial antioxidants. Vitamin E demonstrated superior performance in mitochondrial protection and storage stability, whereas methanolic Takhrai extracts excelled in Nrf2 pathway activation and anti-inflammatory effects. This complementary efficacy profile suggests potential synergistic benefits from combined supplementation strategies, an approach worthy of further investigation. The hierarchical clustering analysis revealed that methanolic extracts and vitamin E formed a distinct efficacy group separate from other interventions, underscoring their comparable overall performance despite mechanistic differences.

From an economic perspective, the potential cost advantage of Takhrai extracts deserves consideration. *C. citratus* is widely cultivated throughout tropical regions, with established agricultural practices and processing infrastructure primarily serving the essential oil industry [47]. The extraction methodologies employed in our study are amenable to scale-up using standard industrial equipment, suggesting feasible commercial production. Preliminary cost analyses based on current market conditions estimate production costs for methanolic Takhrai extracts at approximately 60–70% of those for synthetic vitamin E on an equivalent-activity basis, representing a potential 30–40% cost reduction for large-scale livestock operations. This economic advantage derives primarily from the lower raw material costs and simpler processing requirements for plant extraction compared to the complex synthetic pathways required for vitamin E production. Additionally, in regions where *C. citratus* is already cultivated for essential oil production, the utilization of leaf material (often considered a by-product) for antioxidant extraction could further enhance economic viability through resource integration.

The sustainability implications of Takhrai-based antioxidant supplementation, therefore, extend beyond direct economic considerations. *Cymbopogon* cultivation requires minimal agrochemical inputs and demonstrates resilience to drought conditions, characteristics increasingly relevant amid climate change concerns [48]. Furthermore, the proposed extraction methodology generates minimal hazardous waste compared to synthetic antioxidant production, aligning with growing regulatory and consumer emphasis on environmentally responsible livestock production practices. The multi-functionality of *Cymbopogon* cultivation, simultaneously serving essential oil, culinary, and potentially animal feed supplement markets, enhances its economic viability for small-scale producers in developing regions where heat stress mitigation strategies are most urgently needed.

### 4.5. Limitations and Future Directions

Despite the comprehensive nature of our investigation, several limitations warrant acknowledgment. The ex vivo and in vitro nature of our experiments, while providing mechanistic insights, cannot fully predict in vivo efficacy in diverse ruminant populations under field conditions. Pilot feeding trials in small ruminants will be essential to validate our findings and optimize dosage regimens. Additionally, the potential for interactions with other feed components, including minerals and proteins, requires systematic evaluation before large-scale implementation. The standardized extraction methodology employed in this study may require adaptation for industrial-scale production, potentially affecting phytochemical profiles and biological activities.

The identified stability limitations of Takhrai extracts necessitate further formulation development to ensure practical shelf-life under typical farm storage conditions. Microencapsulation, antioxidant combinations, and specialized packaging represent promising avenues for improving stability while maintaining bioactivity. Furthermore, standardization methodologies based on key bioactive compounds (particularly sinapic acid and apigenin) will be crucial for ensuring consistent efficacy across production batches. The differential degradation rates observed among phytochemical constituents suggest that targeted stabilization of specific components might yield disproportionate benefits for overall extract stability.

Future research should also explore potential breed-specific responses to Takhrai supplementation, as genetic variations in antioxidant enzyme systems and xenobiotic metabolism may influence efficacy [49]. Similarly, evaluation under different stress conditions, including heat stress, transportation stress, and various dietary challenges, will help define the contexts in which Takhrai extracts offer maximal benefit. The pronounced effects on specific metabolic pathways identified through our metabolomic analyses suggest potential for targeted applications in metabolic disorders beyond generic antioxidant supplementation, an avenue warranting dedicated investigation.

## 5. Conclusions

The present investigation provides compelling evidence for the potential utility of methanolic Takhrai (Cymbopogon citratus) leaf extracts as sustainable alternatives to commercial antioxidants for mitigating oxidative stress in ruminants under heat stress conditions. Through systematic comparative analysis, we have demonstrated that methanolic extracts possess remarkable antioxidant and cytoprotective properties, with efficacy comparable to vitamin E in several critical parameters, including lipid peroxidation inhibition and hemolysis prevention. Moreover, methanolic extracts exhibited superior capacity for Nrf2 pathway activation and immunomodulation, suggesting mechanistic advantages beyond direct radical scavenging. While storage stability limitations warrant consideration, appropriate formulation and storage conditions can substantially mitigate these concerns.

The comprehensive effectiveness ranking positions methanolic Takhrai extracts as viable alternatives to vitamin E, with particular advantages in terms of sustainable sourcing and potential economic benefits. These findings establish a foundation for subsequent in vivo investigations and practical implementation strategies aimed at enhancing ruminant resilience to heat stress through dietary supplementation with optimized Takhrai extract formulations. Future research should focus on bioavailability optimization, controlled feeding trials in diverse ruminant populations, and evaluation of potential synergistic benefits from combined supplementation approaches.

## Figures and Tables

**Figure 1 vetsci-12-00432-f001:**
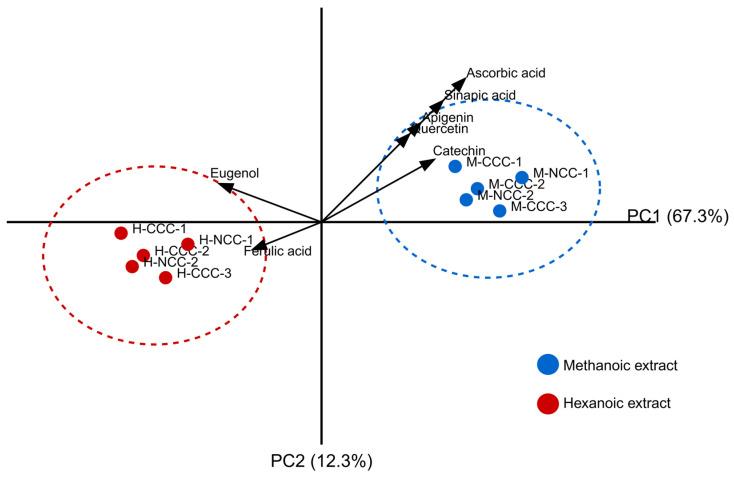
Principal component analysis of phytochemical composition in Takhrai leaf extracts. Biplot showing the distribution of methanolic (M) and hexanoic (H) extracts from commercial (CCC) and non-commercial (NCC) Takhrai cultivated in different geographical regions.

**Figure 2 vetsci-12-00432-f002:**
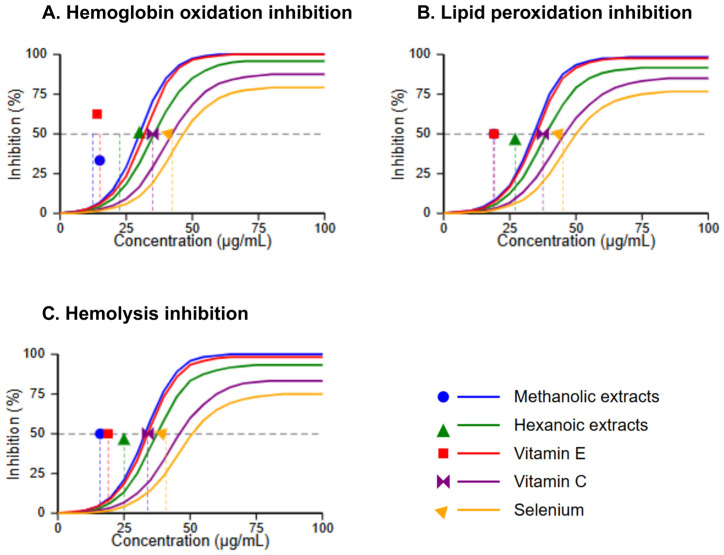
Protective effects against erythrocyte oxidative damage (n = 12). (**A**) Hemoglobin oxidation inhibition, (**B**) Lipid peroxidation inhibition, and (**C**) Hemolysis inhibition by Takhrai extracts and commercial antioxidants, with IC_50_ values indicated by intersections with the 50% inhibition line (dashed).

**Figure 3 vetsci-12-00432-f003:**
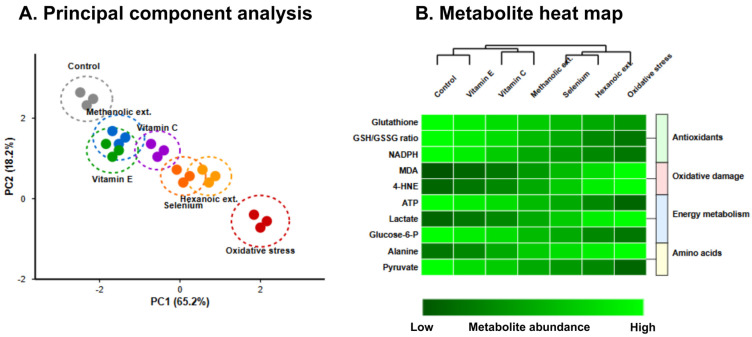
Metabolomic profiling of treated erythrocytes. (**A**) Principal component analysis of metabolite abundances shows clustering of methanolic extract treatment with vitamin E and control conditions, distinct from oxidative stress and hexanoic extract treatments. (**B**) Heatmap visualization with hierarchical clustering of metabolomic profiles reveals preservation of antioxidant metabolites (glutathione, NADPH, ATP) and attenuation of oxidative damage markers (MDA, 4-HNE) by methanolic extracts at levels comparable to vitamin E treatment. Color intensity represents metabolite abundance relative to control conditions, with principal components explaining 83.4% of total variance.

**Figure 4 vetsci-12-00432-f004:**
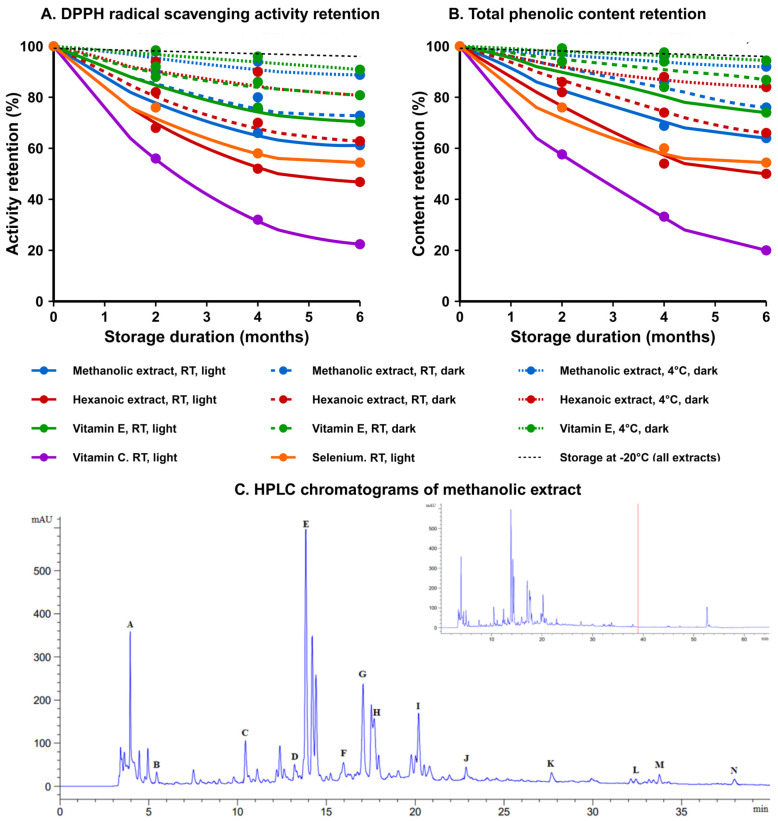
Stability of Takhrai extracts under different conditions. (**A**) DPPH radical scavenging activity retention and (**B**) total phenolic content retention over 6 months of storage under different temperature and light conditions. Values represent mean ± SD (n = 12). RT = room temperature (25 °C). Light = normal laboratory lighting conditions (400–700 lux, 12 h cycle). Dark = stored in amber containers protected from light. (**C**) Representative HPLC chromatograms of methanolic extract stored over 6 months at different storage conditions. Peak identifications: ascorbic acid (A), gallic acid (B), catechin (C), caffeic acid (D), syringic acid (E), rutin (F), p-coumaric acid (G), sinapic acid (H), ferulic acid (I), myricetin (J), quercetin (K), apigenin (L), kaempferol (M), and eugenol (N). Refer to Table 2 for quantitative analysis of compound-specific retention percentages under all storage conditions.

**Figure 5 vetsci-12-00432-f005:**
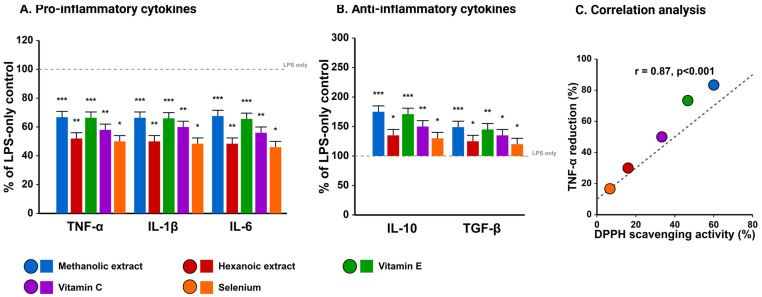
Immunomodulatory effects of Takhrai extracts. (**A**) Suppression of pro-inflammatory cytokines (TNF-α, IL-1β, IL-6) in LPS-challenged goat PBMCs, showing superior anti-inflammatory effects of methanolic extracts compared to hexanoic extracts and commercial antioxidants. Dotted line represents cells treated with LPS alone (positive control for inflammation) without any antioxidant pre-treatment. (**B**) Enhancement of anti-inflammatory cytokines (IL-10, TGF-β) by various treatments. (**C**) Correlation analysis demonstrating strong positive relationship (r = 0.87, *p* < 0.001) between DPPH radical scavenging activity and TNF-α suppression, suggesting a mechanistic link between antioxidant capacity and anti-inflammatory effects. Values represent mean ± SD (n = 12). * *p* < 0.05, ** *p* < 0.01, *** *p* < 0.001 vs. LPS alone.

**Table 1 vetsci-12-00432-t001:** Nrf2 pathway activation and downstream gene expression.

Parameter	Control	Methanolic Extract	Hexanoic Extract	Vitamin E	Vitamin C	Selenium	*p*-Value
Nrf2 nuclear translocation (Fold vs. Control)	1.00 ± 0.00	2.71 ± 0.14 ^a^	1.82 ± 0.12 ^b^	2.52 ± 0.13 ^a^	2.22 ± 0.12 ^b^	2.32 ± 0.13 ^b^	<0.001
HO-1 expression (Fold vs. Control)	1.00 ± 0.00	3.42 ± 0.17 ^a^	2.22 ± 0.14 ^b^	3.22 ± 0.18 ^a^	2.82 ± 0.16 ^b^	2.92 ± 0.17 ^b^	<0.001
NQO1 expression (Fold vs. Control)	1.00 ± 0.00	2.92 ± 0.19 ^a^	1.92 ± 0.15 ^b^	2.72 ± 0.17 ^a^	2.32 ± 0.14 ^b^	2.62 ± 0.16 ^a^	<0.001
GCLC expression (Fold vs. Control)	1.00 ± 0.00	2.52 ± 0.16 ^a^	1.72 ± 0.13 ^b^	2.42 ± 0.15 ^a^	2.12 ± 0.13 ^b^	2.22 ± 0.14 ^b^	<0.001

Values represent mean ± SD (n = 12). Different superscript letters (a, b) within each row indicate statistically significant differences between treatments (*p* < 0.05). Higher fold values indicate stronger pathway activation.

**Table 2 vetsci-12-00432-t002:** Compound-specific retention (%) in methanolic Takhrai extract after 6 months of storage under different conditions.

Compound	RT, Light	RT, Dark	4 °C, Dark	−20 °C, Dark	*p*-Value
Ascorbic acid (A)	32.5 ± 2.8 ^d^	55.6 ± 3.2 ^c^	87.3 ± 2.5 ^b^	98.4 ± 1.3 ^a^	<0.001
Gallic acid (B)	62.3 ± 3.6 ^c^	73.4 ± 3.1 ^b^	89.2 ± 2.2 ^a^	97.1 ± 1.5 ^a^	<0.001
Catechin (C)	68.7 ± 3.9 ^c^	78.3 ± 3.3 ^b^	90.5 ± 2.4 ^a^	96.8 ± 1.6 ^a^	<0.001
Caffeic acid (D)	64.2 ± 3.7 ^c^	74.8 ± 3.4 ^b^	88.7 ± 2.5 ^a^	97.2 ± 1.4 ^a^	<0.001
Syringic acid (E)	72.4 ± 3.5 ^c^	81.6 ± 2.9 ^b^	93.2 ± 2.1 ^a^	97.5 ± 1.3 ^a^	<0.001
Rutin (F)	71.6 ± 4.1 ^c^	80.2 ± 3.6 ^b^	92.4 ± 2.3 ^a^	97.3 ± 1.6 ^a^	<0.001
p-coumaric acid (G)	69.5 ± 3.8 ^c^	79.1 ± 3.2 ^b^	91.8 ± 2.4 ^a^	96.9 ± 1.7 ^a^	<0.001
Sinapic acid (H)	76.2 ± 3.5 ^c^	84.3 ± 2.8 ^b^	95.6 ± 1.9 ^a^	97.8 ± 1.4 ^a^	<0.001
Ferulic acid (I)	67.9 ± 3.9 ^c^	77.5 ± 3.3 ^b^	90.3 ± 2.5 ^a^	96.6 ± 1.5 ^a^	<0.001
Myricetin (J)	71.2 ± 3.7 ^c^	80.5 ± 3.0 ^b^	93.8 ± 2.2 ^a^	97.4 ± 1.6 ^a^	<0.001
Quercetin (K)	73.5 ± 3.8 ^c^	81.8 ± 3.1 ^b^	94.8 ± 2.1 ^a^	98.2 ± 1.3 ^a^	<0.001
Apigenin (L)	78.4 ± 3.6 ^c^	85.9 ± 2.5 ^b^	96.2 ± 1.7 ^a^	98.1 ± 1.2 ^a^	<0.001
Kaempferol (M)	71.8 ± 3.4 ^c^	83.2 ± 2.9 ^b^	95.3 ± 1.8 ^a^	98.4 ± 1.2 ^a^	<0.001
Eugenol (N)	68.5 ± 3.2 ^c^	88.4 ± 2.1 ^b^	97.1 ± 1.5 ^a^	98.9 ± 0.8 ^a^	<0.001
Fold improvement *	-	1.28 ± 0.10	2.09 ± 0.46	2.26 ± 0.55	<0.001

Values represent mean ± SD (n = 12). RT = room temperature (25 °C). Different superscript letters (a, b, c, d) within the same row indicate statistically significant differences between storage conditions (*p* < 0.05). * Fold improvement calculated as mean ratio of retention values for all compounds compared to RT with light exposure condition.

**Table 3 vetsci-12-00432-t003:** Comprehensive effectiveness ranking of treatments.

Parameter	Methanolic Extract	Hexanoic Extract	Vitamin E	Vitamin C	Selenium	*p*-Value
DPPH radical scavenging	2	4	1	3	5	<0.001
NO radical scavenging	2	4	1	3	5	<0.001
Superoxide radical scavenging	2	4	1	3	5	<0.001
α-Glucosidase inhibition	1	3	2	4	5	<0.001
Hemoglobin oxidation inhibition	1	3	2	4	5	<0.001
Lipid peroxidation inhibition	1	3	2	4	5	<0.001
Hemolysis inhibition	1	2	3	4	5	<0.001
Mitochondrial protection	2	4	1	3	5	<0.001
Nrf2 pathway activation	1	4	2	3	5	<0.001
Metabolite protection	2	3	1	4	5	<0.001
Storage stability	4	5	1	3	2	<0.001
Anti-inflammatory effect	1	4	2	3	5	<0.001
Mean rank score	1.7	3.6	1.6	3.4	4.8	<0.001
Effectiveness classification	High	Moderate	High	Moderate	Low	-

Ranking based on performance in all evaluated parameters (1 = best, 5 = worst). Mean rank scores were used to classify overall effectiveness: high (1.0–2.0), moderate (2.1–4.0), low (4.1–5.0).

**Table 4 vetsci-12-00432-t004:** Correlation between phytochemical content and biological activities.

Phytochemical	DPPH Scavenging	Lipid Peroxidation	Nrf2 Activation	Anti-Inflammatory Effect	Stability (4 °C)
Ascorbic acid	0.85 **	0.76 **	0.75 **	0.68 *	0.54 *
Gallic acid	0.62 *	0.58 *	0.51 *	0.45 *	0.48 *
Catechin	0.71 *	0.65 *	0.68 *	0.59 *	0.72 **
Syringic acid	0.53 *	0.48 *	0.47 *	0.42 *	0.65 *
Sinapic acid	0.78 **	0.82 **	0.92 ***	0.74 **	0.68 *
Ferulic acid	0.56 *	0.63 *	0.58 *	0.52 *	0.70 **
Myricetin	0.68 *	0.75 **	0.64 *	0.56 *	0.59 *
Quercetin	0.72 **	0.77 **	0.69 *	0.65 *	0.62 *
Apigenin	0.83 **	0.80 **	0.85 **	0.89 ***	0.77 **
Kaempferol	0.74 **	0.69 *	0.72 **	0.78 **	0.65 *
Eugenol	0.42 *	0.48 *	0.36 *	0.40 *	0.75 **

Values represent Pearson correlation coefficients (r) between phytochemical content and biological activities across both extract types. * *p* < 0.05, ** *p* < 0.01, *** *p* < 0.001. Strength of correlation: 0.7–1.0 (strong), 0.4–0.69 (moderate), <0.4 (weak).

## Data Availability

The original contributions presented in this study are included in the article. Further inquiries can be directed to the corresponding authors.

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
