# Peer review of "Comparative Evaluation of Takhrai (*Cymbopogon citratus*) Leaf Extracts with Commercial Antioxidants for Oxidative Stress Mitigation in Ruminants Under Heat Stress"

_vetsci, 2025, doi:10.3390/vetsci12050432_

Round 1

Reviewer 1 Report

Comments and Suggestions for Authors

 Review: "Comparative Evaluation of Takhrai (Cymbopogon citratus) Leaf Extracts with Commercial Antioxidants for Oxidative Stress 3 Mitigation in Ruminants Under Heat Stress"

The search for new alternatives for the use of antioxidants in ruminant production systems is important in the context of improving sustainability, a point that lends relevance to this research. Furthermore, the quality of this paper is highlighted in terms of the presentation of materials and methods, results, and discussion. Below are some suggestions and recommendations that would improve the manuscript: 

Line 19: When starting the description of results, mention metabolic extracts but does not specify which compound it refers to, except below (line 25) where it is understood that it is talking about Takhrai leaves.

Line 24: Regarding the last paragraph, what is the relationship between cymbopogon cultivation, ruminant production and Takhrai?

The simple summary is somewhat confusing, it is suggested to improve it to make it more explanatory

Line 48: If the concept “methabolic extracts” if mentioned so much, why not are a keyword?

Line 76: what mean “(DC.)” if are the acronym of Cymbopogon citratus why not used later?

Line 102: This study seeks to evaluate and compare the antioxidant effects of a natural compound for use in ruminant production under stressful conditions. The introduction mentions that this product has two extracts, one polar and one nonpolar, but provides no additional information on the characteristics of each. Therefore, it is not clear to me whether the proposed hypothesis emanates from the state of the art associated with the concepts addressed in this research or if it is formulated based on the results obtained in this study, which would be closer to a conclusion.

Introduction: Must be improved. It should provide more information about the compound studied and its extracts given the hypothesis that is raised

Line 120: respect sample collect on consecutive months (March to August) could you indicate the  year?

Material and methods: The methodologies applied are described in detail and the protocols that require it are referenced. The methodology presented is robust and comprehensive, which determines the obtaining of a large data of results.

Results

Line 639: table 1: It is recommended to read the results in the table horizontally, that is, to arrange the treatments and p-value in the rows to read and compare the statistical differences on the horizontal axis.

Figure 4A-B: It is then understood that the best method of conservation of the extracts is at 4°C in darkness for both methanolic and hexanoic extracts and they are compared with vit E at RT and light exposure, why was a vit E control not used under conditions of 4°C and in darkness? Probably this control would achieve better responses in terms of retention of antioxidant activity post storage.

Figure 4C: Why is HPLC performed on a methanolic extract under RT conditions and light exposure and the extract that resulted in better post-storage responses (methanolic at 4°C and in darkness) not presented?

Line 673: If IL-6 results are presented in the text, why are they not shown in the graph in Figure 5A together with TNF-a and IL-1b?

Figure 5: The dotted line LPS-only is somewhat confusing. Could an explanation or meaning be added to the figure caption? Is it the same as LPS alone?

Discussion: It is addressed based on each point analyzed, which provides greater clarity and quality to the writing. The final points address two important topics for research: practical considerations for application in livestock farming, as well as its effectiveness and limitations and future directions, which is noteworthy and valuable.

Conclusion: The results are conclusive and allow for the delivery of a correct and decisive conclusion.

Lines 758-762 are very similar to lines 774-776, It would be advisable to adjust the texts without losing the ideas so that they do not look like a copy

Line 825- line 829: It is not clear what the relationship would be between the effective concentrations (25-100 ug/ml) and the practical recommendations for incorporation (150-250 mg/kg). Could this be explained in a better way?

Line 858: It is indicated that the synthesis of methanolic extracts is 60-70% more expensive than the synthesis of commercial antioxidants of Vit E, so the potential economic advantage associated with large-scale production is taken by the use of Vit E?

Author Response

Dear Reviewer 1

Please find out our responses in attachment below

Best regards, 
Corresponding authors

Reviewer 2 Report

Comments and Suggestions for Authors

This is a very good study. Plant extracts are one of the important means to improve the health and production performance of livestock, but there is limited research and application in ruminants. This study focuses on this and systematically investigates the effects of Cymbopogon citratus leaf extract on ruminants, providing guidance for the application of this extract in ruminants. However, there are some issues with the paper. Firstly, the author's introduction does not highlight the novelty of this study, especially the insufficient description of the research progress in this area, which cannot demonstrate the importance of this study, especially in terms of application value. The author is requested to further supplement and improve this part. Secondly, the author should draw a mechanism diagram of the obtained results, which can help readers understand the results and conclusions of this study very clearly. In summary, the overall quality of this study is good, and after the author's revisions, the editorial department may consider publishing it.

Author Response

Dear Reviewer 2

Please find out our responses in attachment below

Best regards, 
Corresponding authors

Round 2

Reviewer 2 Report

Comments and Suggestions for Authors

The author has made modifications.